# Long Non-Coding RNA Signatures in the Ileum and Colon of Crohn’s Disease Patients and Effect of Anti-TNF-α Treatment on Their Modulation

**DOI:** 10.3390/ijms242115691

**Published:** 2023-10-28

**Authors:** Montse Baldan-Martin, Cristina Rubín de Célix, Macarena Orejudo, Lorena Ortega Moreno, Samuel Fernández-Tomé, Irene Soleto, Cristina Ramirez, Ricardo Arroyo, Paloma Fernández, Cecilio Santander, José Andrés Moreno-Monteagudo, María José Casanova, Fernando Casals, Sergio Casabona, Irene Becerro, Juan J. Lozano, Ana M. Aransay, María Chaparro, Javier P. Gisbert

**Affiliations:** 1Gastroenterology Unit, Hospital Universitario de La Princesa, Instituto de Investigación Sanitaria Princesa (IIS-Princesa), Universidad Autónoma de Madrid (UAM), Centro de Investigación Biomédica en Red de Enfermedades Hepáticas y Digestivas (CIBERehd), 28006 Madrid, Spain; cristina.rubin.92@hotmail.com (C.R.d.C.); macaorejudo@gmail.com (M.O.); irenesoletof@gmail.com (I.S.); ramicristina460@gmail.com (C.R.); cecilio.santander@salud.madrid.org (C.S.); jamorenomonteagudo@hotmail.com (J.A.M.-M.); mjcasanova.g@gmail.com (M.J.C.); drcasals@hotmail.com (F.C.); scasabonafrances@gmail.com (S.C.); ibg_chispas@hotmail.com (I.B.); mariachs2005@gmail.com (M.C.); javier.p.gisbert@gmail.com (J.P.G.); 2Área de Farmacología y Nutrición y Bromatología, Grupo de Investigación de Alto Rendimiento en Fisiopatología del Sistema Digestivo URJC: NeuGut-URJC, Departamento Ciencias Básicas de la Salud, Universidad Rey Juan Carlos, 28922 Madrid, Spain; lorena.ortega8317@gmail.com; 3Departamento de Nutrición y Ciencia de los Alimentos, Facultad de Farmacia, Universidad Complutense de Madrid, 28040 Madrid, Spain; fernandeztome.samuel@gmail.com; 4Instituto de Medicina Molecular Aplicada Nemesio Díez (IMMA-ND), Facultad de Medicina, Universidad San Pablo CEU, 28668 Madrid, Spainpaloma.fernandezmartinez@ceu.es (P.F.); 5Bioinformatics Platform, Centro de Investigación Biomédica en Red de Enfermedades Hepáticas y Digestivas (CIBERehd), 08036 Barcelona, Spain; juanjo.lozano@ciberehd.org; 6Genome Analysis Platform, CIC bioGUNE, Basque Research and Technology Alliance (BRTA), Centro de Investigación Biomédica en Red de Enfermedades Hepáticas y Digestivas (CIBERehd), 48160 Derio, Spain; amaransay@cicbiogune.es

**Keywords:** inflammatory bowel disease, Crohn’s disease, biological drugs, anti-TNF-α, long noncoding RNA, lncRNA, RNA seq

## Abstract

Biological therapies only benefit one-third of patients with Crohn’s disease (CD). For this reason, a deeper understanding of the mechanisms by which biologics elicit their effect on intestinal mucosa is needed. Increasing evidence points toward the involvement of long noncoding RNAs (lncRNAs) in the pathogenesis of CD, although their role remains poorly studied. We aimed to characterize lncRNA profiles in the ileum and colon from CD patients and evaluate the effect of anti-TNF-α treatment on their transcription. Terminal ileum and left colon samples from 30 patients (active CD = 10, quiescent CD = 10, and healthy controls (HCs) = 10) were collected for RNA-seq. The patients were classified according to endoscopic activity. Furthermore, biopsies were cultured with infliximab, and their transcriptome was determined by Illumina gene expression array. A total of 678 differentially expressed lncRNAs between the terminal ileum and left colon were identified in HCs, 438 in patients with quiescent CD, and 468 in patients with active CD. Additionally, we identified three new lncRNAs in the ileum associated with CD activity. No differences were observed when comparing the effect of infliximab according to intestinal location, presence of disease (CD vs. HC), and activity (active vs. quiescent). The expression profiles of lncRNAs are associated with the location of intestinal tissue, being very different in the ileum and colon. The presence of CD and disease activity are associated with the differential expression of lncRNAs. No modulatory effect of infliximab has been observed in the lncRNA transcriptome.

## 1. Introduction

Inflammatory bowel diseases (IBDs) are chronic relapsing and remitting inflammatory conditions of the gastrointestinal tract resulting from a complex interaction between genetic susceptibility, the immune response to the gut microbiome, and environmental factors [1]. Crohn’s disease (CD), an IBD subtype, is characterized by discontinuous and transmural intestinal inflammation, most often affecting the terminal ileum and colon. The global burden of CD is increasing worldwide, with an incidence in Europe between 0.4 and 22.8 per 100,000 people per year [2]. Although CD can affect any part of the gastrointestinal tract, in most cases, the terminal ileum and colon are involved. It is characterized by transmural inflammation, the presence of granulomas, fissuring ulceration, and intestinal fibrosis [3].

Long noncoding RNAs (lncRNAs) are a class of RNAs with a length of over 200 nucleotides that are not translated into functional proteins. Previous studies have reported the role of lncRNAs in the regulation of the immune system and inflammatory pathways in IBD [4,5]: several lncRNAs are dysregulated in the inflamed tissues of patients with CD, such as GAS5, MM2, MMP9, and TUG1 [6]. Recently, Braun T et al. showed that HNF1A-AS1 expression was reduced in mucosal biopsies from IBD patients [7]. However, studies that focus on characterizing lncRNA expression in the ileal and colonic mucosa from active and quiescent CD patients, which can provide a mechanistic explanation for how lncRNAs participate in CD pathogenesis, are lacking. A better characterization of the transcriptome landscape in CD will open new avenues in the discovery of novel therapeutic strategies.

Anti-tumor necrosis factor (TNF)-α therapies, such as infliximab, are effective for treating CD. Nonetheless, up to one-third of patients do not respond to the induction, and among the responders, one-third of these patients lose response with time. The causes underlying primary nonresponse to anti-TNF-α agents are unknown [8,9], and this uncertainty is, in part, related to the fact that the mechanism of action of anti-TNF-α therapy in CD is still not well understood [10]. In this respect, the mechanism of action of anti-TNF-α drugs has been addressed from different perspectives; for example, by using ex vivo studies, our group has recently demonstrated that anti-TNF-α drugs modify the migratory capacity of different circulating dendritic cell subsets [11]. However, the effect of anti-TNF-α on lncRNA expression on CD intestinal mucosa has not been studied yet [12,13].

In the present study, unbiased transcriptomic profiling of lncRNA was performed by RNA sequencing to characterize lncRNA signatures in the inflamed and non-inflamed ileum and colon from CD patients and healthy controls (HCs). In addition, the ex vivo effect of anti-TNF-α on intestinal mucosa was evaluated to provide insights into the mechanisms by which biological drugs elicit their effect at the transcriptome level. Our study will provide meaningful insights into the biological structure of the small bowel and colon and will help us to understand whether anti-TNF-α agents exert an effect on the lncRNA transcriptome.

## 2. Results

### 2.1. Characterization of lncRNA Profiles in the Terminal Ileum and Left Colon

In order to define the terminal ileum and left colon lncRNA landscape in CD patients and HCs, we performed a transcriptome-wide differential expression analysis. The RNA-seq analysis identified a total of 678 differentially expressed lncRNAs between terminal ileum and left colon in HCs (373 upregulated and 305 downregulated), 438 that were significant between the terminal ileum and left colon in patients with qCD (253 upregulated and 185 downregulated), and 468 (312 upregulated and 156 downregulated) in patients with aCD (Table 1 and Appendix A).

A Venn diagram illustrating the number of overlapping lncRNAs differentially expressed in HC, qCD, and aCD is shown in Figure 1. Of these, 287 lncRNAs were significant only in HC, 95 only in qCD, and 184 only in aCD.

A principal component analysis (PCA) plot analysis was performed to evaluate the difference in lncRNAs in the terminal ileum vs. the left colon of HC, qCD, and aCD (Figure 2A, 2B and 2C, respectively). The results showed that lncRNA expression profiles could clearly distinguish between the groups. Furthermore, to observe the differences in lncRNA expression between the locations in HC, qCD, and aCD, volcano plots were generated (Figure 3A–C). The cluster heatmap of the top 50 differentially expressed lncRNAs showed significant differences between terminal ileum and left colon in HCs, quiescent, and active CD patients (Figure 4A, 4B, and 4C, respectively).

The top 10 upregulated and downregulated lncRNAs based on FC values for terminal ileum vs. left colon in HCs were the following: HNF4A-AS1, FABP6-AS1, ENSG00000235122, ENSG00000261012, ENSG00000237153, ENSG00000233376, LINC00330, ENSG00000249201, LINC00955, APOA1-AS, HOTTIP, ENSG00000228222, HOXA11-AS, LINC02023, ENSG00000231412, LINC01285, SATB2-AS1, ENSG00000229261, VLDLR-AS1, and ENSG00000225421 (Table 2). The top 10 upregulated and downregulated lncRNAs in the terminal ileum vs. left colon comparisons in qCD patients included HNF4A-AS1, FABP6-AS1, ENSG00000261012, ENSG00000235122, APOA1-AS, ENSG00000237153, MIR31HG, ENSG00000233376, ENSG00000260597, EGFR-AS1, HOTTIP, LINC02023, SATB2-AS1, ENSG00000254645, ENSG00000225421, HOXA11-AS, ENSG00000231412, ENSG00000228222, LINC01285, and ENSG00000229261 (Table 3). Finally, the 10 lncRNAs most up/downregulated lncRNAs between the terminal ileum and left colon in aCD patients were HNF4A-AS1, FABP6-AS1, ENSG00000235122, ENSG00000260597, ENSG00000261012, ENSG00000249201, LINC02404, MIR31HG, APOA1-AS, ENSG00000233376, HOTTIP, LINC02023, SATB2-AS1, ENSG00000228222, HOXA11-AS, ENSG00000225421, LINC02441, LINC02568, ENSG00000254645, and LINC00520 (Table 4). Regarding location-specific (ileum and colon) and disease-independent (HC, qCD, and aCD) lncRNAs, 182 lncRNAs overlapping between three study comparisons were identified (Appendix A). In this study, we provide a comprehensive transcriptomic view of the differentially expressed lncRNAs between the terminal ileum and the left colon.

### 2.2. Identification of Differential lncRNAs Associated with Disease Activity in the Terminal Ileum of CD Patients

In order to identify differentially expressed lncRNAs in the inflamed tissue of CD patients, we compared the expression profiles of the terminal ileum and left colon samples from aCD patients with HCs. Our results identified three lncRNAs associated with the inflamed ileum from aCD patients compared with the non-inflamed terminal ileum from HCs: LINC02390, ENSG00000257764, and ENSG00000254802 (Table 5). When we compared lncRNA expression between the inflamed left colon and non-inflamed left colon from aCD and HCs, respectively, no significant differences were observed.

### 2.3. Effect of Infliximab on lncRNA Expression in Intestinal Tissue

For a better understanding of the regulation of lncRNA expression in the terminal ileum and left colon of aCD, qCD, and HCs after infliximab culture, we compared the following six study groups: (1) HC: terminal ileum (basal) vs. terminal ileum (infliximab); (2) HC: colon (basal) vs. colon (infliximab); (3) qCD: terminal ileum (basal) vs. terminal ileum (infliximab); (4) qCD: left colon (basal) vs. left colon (infliximab); (5) aCD: terminal ileum (basal) vs. terminal ileum (infliximab); (6) aCD: left colon (basal) vs. left colon (infliximab). No differences were identified for the lncRNA expression profile in the aforementioned comparisons (with vs. without infliximab in culture).

## 3. Discussion

Our results revealed that most of the differences in lncRNA expression in intestinal tissue are driven by location (terminal ileum vs. left colon). These results underscore the importance of intestinal tissue location (terminal ileum or left colon) for lncRNA expression, suggesting compartmentalization of the gastrointestinal tract at the transcriptome level. On the contrary, we could not identify any modulatory effect of infliximab on lncRNA expression.

In this sense, the transcriptomic analyses using RNA-seq revealed the location-specific transcription patterns of lncRNAs. The data showed that 678, 438, and 468 lncRNAs were differentially expressed between the terminal ileum and the left colon in HCs, qCD, and aCD, respectively. It is worth mentioning that most differentially expressed lncRNAs are uncharacterized and have not been studied yet. Venn diagrams displayed 182 lncRNAs that were differentially expressed in the aforementioned three study comparisons, indicating that these lncRNAs are exclusively location-specific and not due to the presence of disease or activity. Using the ToppGene Suite tool, we only found biological information of four lncRNAs identified (DELEC1, CRYZL2P-SEC16B, HOTTIP, and HYMAI). DELEC1 (Deleted in esophageal cancer 1) acts as a negative regulator of cell proliferation [14], CRYZL2P-SEC16B (CRYZL2P-SEC16B readthrough) is located in the endoplasmic reticulum exit site, and transport vesicle membrane and it is mainly involved in the regulation of intracellular protein transport, peroxisome fission, and vesicle targeting. The main biological functions of HOTTIP (HOXA distal transcript antisense RNA) include skeletal muscle fiber development, myotube cell development, and neuromuscular processes. Finally, HYMAI (hydatidiform mole associated and imprinted) plays an important role in DNA alkylation and methylation.

Moreover, our analysis identified 184 differentially expressed lncRNAs exclusively in the comparison between the terminal ileum and left colon in aCD patients. This finding might indicate a possible role for those lncRNAs in the pathogenesis of the disease. Of these 184 lncRNAs, we only found information on biological functions in four (FIRRE, LINC00987, DUBR, and DIO3OS). FIRRE (functional intergenic repeating RNA element), a nuclear-retained and chromatin-associated lncRNA, regulates the expression of several inflammatory genes in macrophages and intestinal epithelial cells [15]. In our study, the expression levels of FIRRE were downregulated in the ileum compared to the colon. LINC00987 (long intergenic nonprotein coding RNA 987) was upregulated in colonic biopsies. A previous study has demonstrated that LINC00987 protects against lipopolysaccharide-induced apoptosis, oxidative stress, inflammation, and autophagy by regulating let-7b-5p/SIRT1 axis in bronchial epithelial cells [16]. Another differential lncRNA in intestinal tissue from aCD was DUBR (DPPA2 upstream binding RNA), which is overexpressed in the left colon. DUBR has been related to the positive regulation of myoblast differentiation, the regulation of DNA methylation and alkylation, and the regulation of DNA metabolic processes [17]. Moreover, our data revealed the downregulation of DIO3OS (DIO3 opposite strand upstream RNA) in the left colon samples from aCD when compared to the ileum samples from the same patients. DIO3OS is specifically involved in the immune system, and its expression is differential in several cancer types [18]. Previous studies have described low expression of DIO3OS in colon biopsies from CD patients compared with those from HCs [19,20].

When lncRNA expression in the ileum was compared between aCD patients and HCs, three transcripts with as yet unknown functions were differentially expressed (ENSG00000256582, ENSG00000257764, and ENSG00000254802). A deeper understanding of the function of these lncRNAs is required in future studies to elucidate the molecular mechanisms underlying CD pathogenesis.

Regarding the ex vivo study to evaluate the modulating effect of infliximab on lncRNA expression in intestinal biopsies, our study did not show a regulatory effect of anti-TNF-α on lncRNA transcription on intestinal tissue after culture, according to intestinal location, presence of disease, or activity.

In this study, the specific lncRNA expression patterns in the terminal ileum and left colon may provide a basis for a greater understanding of the molecular mechanisms that drive ileal and colonic CD, which could help to understand different responses to drugs in ileal or colonic CD. Moreover, previous studies have also indicated the significant heterogeneity in ileal and colonic CD, suggesting that colonic-predominant CD is much more closely related to UC and more distinct from ileal CD [21,22]. Of note, few of the differentially expressed lncRNAs identified in our study had been previously described (mainly in the context of other diseases); the majority of them have been described in this work for the first time.

The present work has several limitations. First, the RNA-Seq data were limited to a relatively small number of samples (10 aCD, 10 qCD, and 10 HC). Nevertheless, it is important to note that each patient was studied in both conditions (baseline and after infliximab culture). Secondly, the described functions and potential mechanisms of action of the lncRNAs identified are still not fully elucidated. Accordingly, future in vitro studies are necessary to understand the precise role of these lncRNAs in the pathogenesis of CD.

To our knowledge, this is the first study that analyzed lncRNA expression at the transcript level in the terminal ileum and left colon from patients with CD (active and quiescent) and HCs cultured ex vivo with infliximab. It is worth mentioning that the vast majority of lncRNAs are new transcripts that have not yet been characterized. Maybe this is because, until recently, they had not been given much importance, and furthermore, only recently has the use of Illumina-based RNA sequencing has enabled greater sensitivity and accuracy to detect large numbers of lncRNAs. In addition, the majority of data on the function of lncRNAs come from oncology studies, and their role in other diseases has not been well studied. Therefore, the understanding of the role of lncRNAs in different biological processes and their molecular mechanisms is very limited.

In conclusion, our results provide an in-depth analysis of the lncRNA landscape in the ileum and colon of patients with CD (active and quiescent) and HCs. In addition, the level of expression of some transcripts was associated with the presence of inflammation in CD. Given the lack of knowledge of the role of lncRNAs in the pathogenesis of CD, their location-specific expression profile might lay a foundation for future research to investigate potential therapeutic targets in ileal and colonic CD.

## 4. Materials and Methods

### 4.1. Sample Collection from Patients and Controls

All the patient samples were collected at the Gastroenterology Department of Hospital Universitario de La Princesa (Madrid, Spain). In total, 120 terminal ileum and left colon biopsy samples were obtained from 10 patients with CD with endoscopic activity (aCD), 10 patients with CD in endoscopic remission (qCD), and 10 healthy controls (HCs). The eligibility criteria included patients with CD who had attended a colonoscopy with sedation performed according to medical criteria. Patients were classified using the Simple Endoscopic Score for Crohn’s Disease (SES-CD) as aCD (SES-CD ≥ 3) or qCD (SES-CD  ≤  2). The exclusion criteria included an age of less than 18 years, chronic disease or any other advanced clinically significant pathology, patients who have received anti-TNF, anti-p40, anti-alpha 4 beta 7, or other drugs that may alter the immune system (if taking another nonbiological medication, this must have remained unchanged in the 3 months prior to colonoscopy), as well as alcohol or drugs and pregnancy or lactation. Patients referred due to changes in bowel transit, colorectal cancer screening, or rectal bleeding and without known inflammatory, autoimmune, or malignant diseases were included as HCs; all of them had macroscopically and histologically normal mucosa. Exclusion criteria for HCs included an age of less than 18 years, chronic disease or any other advanced clinically significant pathology, unchanged medication in the 3 months prior to colonoscopy, alcohol or drugs, and pregnancy or lactation. This study was approved by the local Ethics Committee of Hospital Universitario de La Princesa in Madrid, Spain (Protocol GIS-INH-2015; date of approval: 5 October 2017), and written informed consent was obtained from all the participants prior to sample collection. The main demographic characteristics of the CD patients (aCD and qCD) and HCs are shown in Table 6.

### 4.2. Ex Vivo Treatment with Infliximab

Biopsies of inflamed and non-inflamed terminal ileum and left colon from CD patients and HCs were collected in sterile vials and preserved in ice-cold complete culture medium [RPMI 1640 (Sigma-Aldrich, St. Louis, MO, USA), with 100 μg/mL penicillin/streptomycin, 2 mM L-glutamine, 50 μg/mL gentamicin (Sigma-Aldrich), and 10% fetal bovine serum (TCS cellworks)]. All samples were processed within a maximum of 30 min after collection. From each location, explants were cultured in complete medium in a 24-well cell culture plate at 37 °C in the presence of infliximab (10 μg/mL). In parallel, a control culture without the drug was carried out. After 18 h of culture, biopsies were placed in 250 μL RNAlater solution (QIAGEN, Hilden, Germany), frozen at −80 °C, and stored for later use.

### 4.3. RNA Extraction and Analysis of lncRNA Expression

Intestinal tissues suspended in TRIZOL Reagent (Tri Reagent; Invitrogen, Waltham, MA, USA) and chloroform (Panreac) were homogenized and subjected to RNA isolation with the RNeasy Mini Kit 250 according to the manufacturer’s instructions. The quantity and quality of the RNAs were evaluated using Qubit RNA HS Assay Kit (Thermo Fisher Scientific, Cat.#Q32855, Waltham, MA, USA) and Agilent RNA 6000 Nano Chips (Agilent Technologies, Cat.# 5067-1511), respectively. Sequencing libraries were prepared using “Illumina^®^ Stranded Total RNA Prep, Ligation with Ribo-Zero Plus” kit (Illumina, Inc., San Diego, CA, USA) following the “Illumina Stranded total RNA Pep, Ligation with Ribo-Zero Plus Reference Guide”. Starting from 58–300 ng of total RNA, rRNA was depleted, and the remaining RNA was purified, fragmented, and primed for cDNA synthesis. cDNA first-strand synthesis was carried out for 10 min at 25 °C, 15 min at 42 °C, 15 min at 70 °C, and paused at 4 °C, and cDNA second strand was synthesized at 16 °C for 1 h. Following A-tailing, the pre-index anchors were ligated to the ends of the double-stranded cDNA fragments to prepare them for dual indexing. A subsequent PCR amplification step to add the index adapter sequences (30 s at 98 °C; 12–15 cycles of 10 s at 98 °C, 30 s at 60 °C, 30 s at 72 °C; 5 min at 72 °C, and paused at 4 °C) was performed. After a final library clean-up, libraries were visualized on an Agilent 2100 Bioanalyzer using Agilent High Sensitivity DNA kit (Agilent Technologies, Cat.#5067-4626), and these were quantified using Qubit ds DNA HS DNA Kit (Thermo Fisher Scientific, Cat.#Q32854). Differential expression of lncRNAs was analyzed using the edgeR package. For all *p*-values, a false discovery rate (FDR) was applied to correct the statistical significance of multiple testing. Transcripts with adjusted *p*-value (FDR) < 0.05 and an absolute fold change (FC) > 2.0 were considered differentially expressed. In addition, we used different databases (Ensembl Biomart, RNAcentral, and ToppGene Suite) to search for non-annotated lncRNA information and data on differentially expressed lncRNAs, including their location (cellular component), involvement in biological processes, and molecular function.

## Figures and Tables

**Figure 1 ijms-24-15691-f001:**
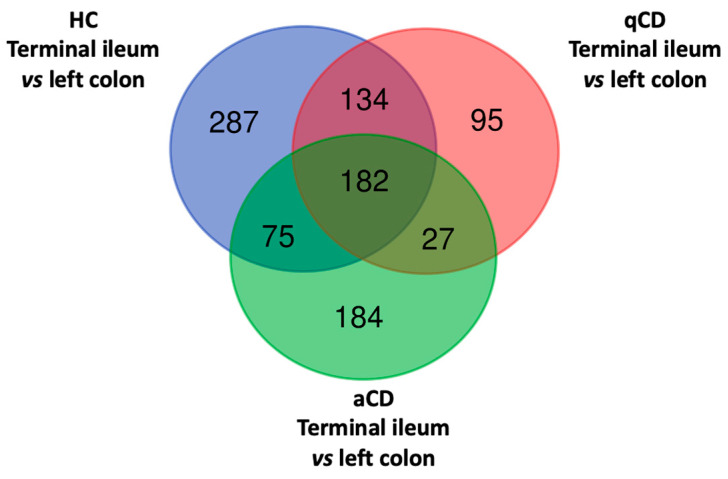
Venn diagram of overlapping differentially expressed lncRNAs between different comparisons. The Venn diagrams show the numbers of lncRNAs differentially expressed between groups (terminal ileum vs. left colon from healthy controls, terminal ileum vs. left colon from quiescent Crohn’s disease [CD] patients, and terminal ileum vs. left colon from active CD patients). aCD: active Crohn’s disease; qCD: quiescent Crohn’s disease; HC: healthy control.

**Figure 2 ijms-24-15691-f002:**
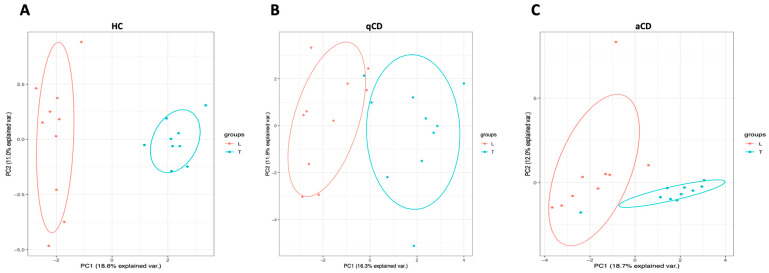
PCA plot analysis of differentially expressed lncRNAs. Comparisons of terminal ileum vs. left colon from healthy controls (**A**), terminal ileum vs. left colon from quiescent Crohn’s disease (CD) patients (**B**), and terminal ileum vs. left colon from active CD patients (**C**). Red and blue dots represent samples of the left colon and terminal ileum, respectively. PCA: principal component analysis; L: left colon; T: terminal ileum; HC: healthy control; qCD: quiescent Crohn’s disease; aCD: active Crohn’s disease.

**Figure 3 ijms-24-15691-f003:**
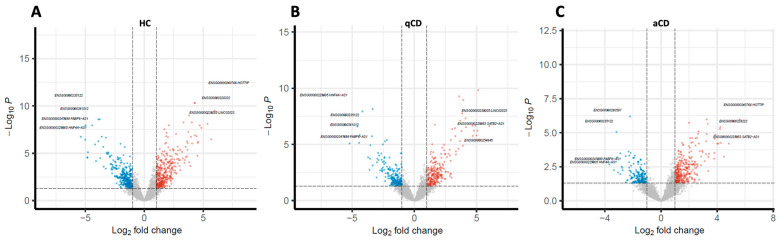
Volcano plots representing fold changes in lncRNA expression. Comparisons of terminal ileum vs. left colon from healthy controls (**A**), terminal ileum vs. left colon from quiescent Crohn’s disease (CD) patients (**B**), and terminal ileum vs. left colon from active CD patients (**C**). Volcano plots show the −log10 (*p*-value) vs. log2 (fold change) of the normalized counts. Red and blue dots represent lncRNAs significantly up or downregulated, respectively. HC: healthy control; qCD: quiescent Crohn’s disease; aCD: active Crohn’s disease.

**Figure 4 ijms-24-15691-f004:**
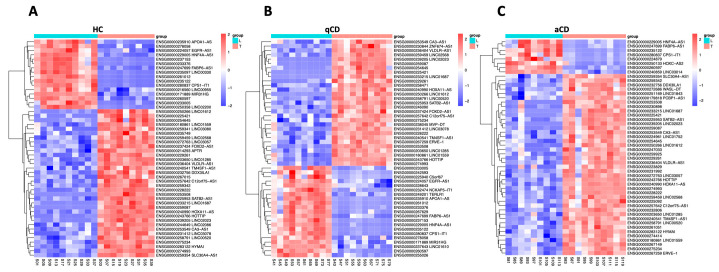
Heatmap analysis of the top 50 differentially expressed lncRNAs. Comparison of terminal ileum vs. left colon from healthy controls (**A**), terminal ileum vs. left colon from quiescent Crohn’s disease (CD) patients (**B**), and terminal ileum vs. left colon from active CD patients (**C**). HC: healthy control; qCD: quiescent Crohn’s disease; aCD: active Crohn’s disease; L: left colon; T: terminal ileum.

**Table 1 ijms-24-15691-t001:** Total number of identified lncRNAs for each group comparison.

	HC Terminal Ileum vs. Left Colon	qCD Terminal Ileum vs. Left Colon	aCD Terminal Ileum vs. Left Colon
lncRNAs downregulated	305	185	156
lncRNAs upregulated	373	253	312
Differential expression of lncRNAs exclusive for each comparison	287	95	184
No significant differences	3088	3328	3298
Total identified lncRNAs	3766	3766	3766

HC: healthy control; qCD: quiescent Crohn’s disease, aCD: active Crohn’s disease.

**Table 2 ijms-24-15691-t002:** Top 10 down- and upregulated lncRNAs, comparing terminal ileum vs. left colon in healthy controls.

Gene Name	Transcript	Chromosome	Differential Expression	Fold Change	*p*-Value	FDR
HNF4A-AS1	ENSG00000229005	20	Downregulated	−119.756	1.45 × 10^−10^	2.04 × 10^−08^
FABP6-AS1	ENSG00000247699	5	Downregulated	−107.655	9.71 × 10^−12^	2.25 × 10^−09^
	ENSG00000235122	6	Downregulated	−84.137	1.07 × 10^−14^	7.54 × 10^−12^
	ENSG00000261012	2	Downregulated	−60.219	4.83 × 10^−13^	1.94 × 10^−10^
	ENSG00000237153	9	Downregulated	−41.887	2.52 × 10^−09^	1.80 × 10^−07^
	ENSG00000233376	20	Downregulated	−32.207	6.32 × 10^−09^	3.65 × 10^−07^
LINC00330	ENSG00000235097	13	Downregulated	−31.710	6.92 × 10^−11^	1.12 × 10^−08^
	ENSG00000249201	5	Downregulated	−28.431	1.44 × 10^−06^	2.63 × 10^−05^
LINC00955	ENSG00000216560	4	Downregulated	−28.400	1.64 × 10^−06^	2.93 × 10^−05^
APOA1-AS	ENSG00000235910	11	Downregulated	−27.507	3.08 × 10^−07^	7.75 × 10^−06^
HOTTIP	ENSG00000243766	7	Upregulated	142.389	9.84 × 10^−17^	3.69 × 10^−13^
	ENSG00000228222	2	Upregulated	68.947	2.12 × 10^−14^	1.40 × 10^−11^
HOXA11-AS	ENSG00000240990	7	Upregulated	62.243	9.04 × 10^−13^	3.23 × 10^−10^
LINC02023	ENSG00000239205	3	Upregulated	52.932	1.62 × 10^−12^	5.28 × 10^−10^
	ENSG00000231412	19	Upregulated	51.174	5.91 × 10^−09^	3.44 × 10^−07^
LINC01285	ENSG00000203650	X	Upregulated	40.996	4.43 × 10^−11^	7.85 × 10^−09^
SATB2-AS1	ENSG00000225953	2	Upregulated	40.760	2.05 × 10^−12^	6.14 × 10^−10^
	ENSG00000229261	10	Upregulated	28.692	1.51 × 10^−10^	2.09 × 10^−08^
VLDLR-AS1	ENSG00000236404	9	Upregulated	26.062	7.02 × 10^−11^	1.12 × 10^−08^
	ENSG00000225421	2	Upregulated	23.790	1.29 × 10^−08^	6.22 × 10^−07^

Results filtered by fold-change. FDR: false discovery rate.

**Table 3 ijms-24-15691-t003:** Top 10 down- and upregulated lncRNAs, comparing terminal ileum vs. left colon in quiescent Crohn’s disease patients.

Gene Name	Transcript	Chromosome	Differential Expression	Fold Change	*p*-Value	FDR
HNF4A-AS1	ENSG00000229005	20	Downregulated	−156.835	2.84 × 10^−13^	4.26 × 10^−10^
FABP6-AS1	ENSG00000247699	5	Downregulated	−50.354	1.86 × 10^−08^	2.00 × 10^−06^
	ENSG00000261012	2	Downregulated	−48.159	9.09 × 10^−10^	1.76 × 10^−07^
	ENSG00000235122	6	Downregulated	−47.026	6.94 × 10^−11^	2.49 × 10^−08^
APOA1-AS	ENSG00000235910	11	Downregulated	−36.773	1.09 × 10^−07^	7.94 × 10^−06^
	ENSG00000237153	9	Downregulated	−23.877	1.49 × 10^−09^	2.53 × 10^−07^
MIR31HG	ENSG00000171889	9	Downregulated	−21.687	9.56 × 10^−08^	7.23 × 10^−06^
	ENSG00000233376	20	Downregulated	−20.921	1.32 × 10^−08^	1.52 × 10^−06^
	ENSG00000260597	12	Downregulated	−18.035	2.41 × 10^−11^	1.15 × 10^−08^
EGFR-AS1	ENSG00000224057	7	Downregulated	−13.122	4.05 × 10^−06^	1.41 × 10^−04^
HOTTIP	ENSG00000243766	7	Upregulated	64.692	1.33 × 10^−09^	2.32 × 10^−07^
LINC02023	ENSG00000239205	3	Upregulated	49.087	1.93 × 10^−11^	9.88 × 10^−09^
SATB2-AS1	ENSG00000225953	2	Upregulated	40.632	7.14 × 10^−10^	1.46 × 10^−07^
	ENSG00000254645	11	Upregulated	36.300	4.67 × 10^−08^	4.18 × 10^−06^
	ENSG00000225421	2	Upregulated	35.127	7.90 × 10^−14^	1.48 × 10^−10^
HOXA11-AS	ENSG00000240990	7	Upregulated	33.482	4.40 × 10^−09^	5.97 × 10^−07^
	ENSG00000231412	19	Upregulated	31.705	1.37 × 10^−08^	1.57 × 10^−06^
	ENSG00000228222	2	Upregulated	29.879	1.10 × 10^−09^	2.08 × 10^−07^
LINC01285	ENSG00000203650	X	Upregulated	27.609	4.32 × 10^−09^	5.93 × 10^−07^
	ENSG00000229261	10	Upregulated	25.831	1.56 × 10^−08^	1.76 × 10^−06^

Results filtered by fold-change. FDR: false discovery rate.

**Table 4 ijms-24-15691-t004:** Top 10 down- and upregulated lncRNAs, comparing ileum vs. left colon in active Crohn’s disease patients.

Gene Name	Transcript	Chromosome	Differential Expression	Fold Change	*p*-Value	FDR
HNF4A-AS1	ENSG00000229005	20	Downregulated	−28.509	5.88 × 10^−05^	1.40 × 10^−03^
FABP6-AS1	ENSG00000247699	5	Downregulated	−22.960	2.81 × 10^−05^	8.57 × 10^−04^
	ENSG00000235122	6	Downregulated	−21.024	3.37 × 10^−09^	1.43 × 10^−06^
	ENSG00000260597	12	Downregulated	−13.933	2.41 × 10^−10^	2.26 × 10^−07^
	ENSG00000261012	2	Downregulated	−12.862	1.13 × 10^−04^	2.12 × 10^−03^
	ENSG00000249201	5	Downregulated	−10.286	4.78 × 10^−04^	5.57 × 10^−03^
LINC02404	ENSG00000257893	12	Downregulated	−9.078	3.30 × 10^−08^	8.74 × 10^−06^
MIR31HG	ENSG00000171889	9	Downregulated	−8.741	1.72 × 10^−05^	6.35 × 10^−04^
APOA1-AS	ENSG00000235910	11	Downregulated	−8.398	8.25 × 10^−04^	8.17 × 10^−03^
	ENSG00000233376	20	Downregulated	−7.475	1.24 × 10^−03^	1.08 × 10^−02^
HOTTIP	ENSG00000243766	7	Upregulated	59.233	4.88 × 10^−11^	9.14 × 10^−08^
LINC02023	ENSG00000239205	3	Upregulated	48.712	2.31 × 10^−09^	1.06 × 10^−06^
SATB2-AS1	ENSG00000225953	2	Upregulated	40.949	1.05 × 10^−07^	2.00 × 10^−05^
	ENSG00000228222	2	Upregulated	36.092	3.60 × 10^−09^	1.45 × 10^−06^
HOXA11-AS	ENSG00000240990	7	Upregulated	35.241	3.37 × 10^−09^	1.43 × 10^−06^
	ENSG00000225421	2	Upregulated	28.643	5.92 × 10^−07^	6.34 × 10^−05^
LINC02441	ENSG00000256643	12	Upregulated	22.430	1.45 × 10^−05^	5.65 × 10^−04^
LINC02568	ENSG00000259459	15	Upregulated	21.576	9.98 × 10^−08^	1.92 × 10^−05^
	ENSG00000254645	11	Upregulated	19.217	4.74 × 10^−07^	5.53 × 10^−05^
LINC00520	ENSG00000258791	14	Upregulated	18.637	1.18 × 10^−08^	3.79 × 10^−06^

Results filtered by fold-change. FDR: false discovery rate.

**Table 5 ijms-24-15691-t005:** Differentially expressed lncRNAs between the terminal ileum from active Crohn’s disease patients and healthy controls.

Gene Name	Transcript	Chromosome	Differential Expression	Fold Change	*p*-Value	FDR
LINC02390	ENSG00000256582	12	Downregulated	−2.530	3.81 × 10^−05^	0.039
	ENSG00000257764	12	Upregulated	4.528	2.83 × 10^−05^	0.039
	ENSG00000254802	8	Upregulated	5.032	1.02 × 10^−05^	0.027

Results filtered by fold-change. FDR: false discovery rate.

**Table 6 ijms-24-15691-t006:** Demographic and clinical characteristics of the study groups.

Characteristics	HC (n = 10)	qCD (n = 10)	aCD (n = 10)
Female gender, n (%)	8 (80)	6 (60)	5 (50)
Hispanic ethnicity, n (%)	10 (100)	10 (100)	10 (100)
Nonsmokers at diagnosis, n (%)	8 (80)	8 (80)	4 (40)
Median age at diagnosis, years (IQR)	N/A	29 (22–37)	27 (22–49)
Median age at enrollment, years (IQR)	51 (45–55)	44 (34–48)	28 (24–57)
CD characteristics	N/A		
Location			
L1: ileal		4 (40)	3 (30)
L2: colic		4 (40)	3 (30)
L3: ileocolic		2 (20)	3 (30)
L4: upper CD		0 (0)	1 (10) *
Behavior			
B1: inflammatory		9 (90)	9 (90)
B2: stricturing		1 (10)	1 (10)
B3: penetrating		0 (0)	0 (0)
Perianal disease		5 (50)	2 (20)
CURRENT TREATMENT **			
None, n (%)	N/A	1 (10)	4 (40)
Aminosalicylates, n (%)		2 (20)	1 (10)
Immunomodulators, n (%)		6 (60)	4 (40)
Biologics, n (%)		4 (40)	4 (40)
Anti-TNF-α agents		4 (40)	1 (10)
1 anti-TNF-α		4 (40)	1 (10)
2 anti-TNF-α		0 (0)	0 (0)
Anti-integrin		0 (0)	2 (20)
Anti-IL-12/23		0 (0)	1 (10)

HC: healthy control; qCD: quiescent Crohn’s disease; aCD: active Crohn’s disease; CD: Crohn’s disease; N/A: not applicable; TNF: tumor necrosis factor; * Ileocolic and upper CD; ** At time of sample collection; IQR, interquartile range.

## Data Availability

All data generated during this study are included in the article or Appendix A.

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
