# Peer review of "Long Non-Coding RNA Signatures in the Ileum and Colon of Crohn’s Disease Patients and Effect of Anti-TNF-α Treatment on Their Modulation"

_ijms, 2023, doi:10.3390/ijms242115691_

Round 1

Reviewer 1 Report

Comments and Suggestions for Authors

I would like to congratulate the work, as it presents new features that can contribute to the diagnosis and treatment of Crohn's Disease. Some points that I highlight below need to be evaluated so that the paper's proposal can be further improved.

1) Introduction: In the first paragraph, it would be worth adding the information that Crohn's Disease is part of Inflammatory Bowel Disease (IBD) and in provide a short explanation about IBD.

2) Material and methods: I believe it is necessary to add the ethics committee approval number, as well as describe the inclusion and exclusion criteria for selecting the work groups, it was not very clear how such selection was made.

3) Results: in item 3.3 I suggest you add a figure/graph or table demonstrating that the values obtained in the ex vivo test, even if they were not significant values, I think it would be valid to represent these results in some way.

4) Discussion: This section is the one that I suggest you pay the most attention to, in the first paragraphs the feeling is that you are only reading the results and not the discussion of these results, as there is practically no comparison with other studies. If there are few studies on lncRNA, use this information when discussing your results as a way to justify the absence of more references. And at some points I felt the lack of reference to the cited excerpt, below are some points that should be referenced:

4.a) lines 249-252

4.b) lines 252-254

4.c) lines 254-255

4.d) line 260 (add the studies about the four IncRNAs described)

4.e) lines 269-271 

Author Response

Response to Reviewer 1 Comments

Point 1. Introduction: In the first paragraph, it would be worth adding the information that Crohn's Disease is part of Inflammatory Bowel Disease (IBD) and in provide a short explanation about IBD.

Response 1: We appreciate the reviewer´s comments. Following the reviewer´s suggestion, a brief explanation of inflammatory bowel diseases we have included in the new version of the manuscript. In addition, we have specified that Crohn´s disease is a subtype of inflammatory bowel disease.

Point 2. Material and methods: I believe it is necessary to add the ethics committee approval number, as well as describe the inclusion and exclusion criteria for selecting the work groups, it was not very clear how such selection was made.

Response 2: We have considered your comments and we have included the ethics committee approval number as well as the criteria for the inclusion and exclusion of subjects.

Point 3. Results: in item 3.3 I suggest you add a figure/graph or table demonstrating that the values obtained in the ex vivo test, even if they were not significant values, I think it would be valid to represent these results in some way.

Response 3: We thank the reviewer for his suggestion. However, we find it difficult to show the list of non-differential lncRNAs in the different study comparisons in a table, as the number of lncRNAs identified was very high (3767 lncRNAs).

Point 4. Discussion: This section is the one that I suggest you pay the most attention to, in the first paragraphs the feeling is that you are only reading the results and not the discussion of these results, as there is practically no comparison with other studies. If there are few studies on lncRNA, use this information when discussing your results as a way to justify the absence of more references. And at some points I felt the lack of reference to the cited excerpt, below are some points that should be referenced:

4.a) lines 249-252

4.b) lines 252-254

4.c) lines 254-255

4.d) line 260 (add the studies about the four IncRNAs described)

4.e) lines 269-271 

Response 4: We agree with the reviewer and as indicated in the discussion " It is worth mentioning that most differentially expressed lncRNAs are uncharacterized and have not been studied yet". Regarding the lack of references in the cited fragments, it is important to mention that, as detailed in the manuscript, information related to the biological function of the identified lncRNAs has been obtained from the website https://toppgene.cchmc.org/. For some lncRNAs, no published studies have been found."

Reviewer 2 Report

Comments and Suggestions for Authors

This is a very sound study in which authors search for lncRNAs from de ileum and left colon of patients with quiescient or active CD. An important result is that data vary considerably from colon to ileum. Also, the authors detect differential expressed lncRNAs in the samples, and then elaborate on possible actions of those that have been previously describe, while other transcripts found are new to literature. Than they repeat assays after incubating biopsy samples in the presence of infliximab, and they find no differences in lncRNA expression. These results are in accordance to that found for patients under infliximab therapy.

Altogether the paper is well written and opens doors to knew studies on CD pathophysiology.

Author Response

We thank the reviewer for his comments.

Reviewer 3 Report

Comments and Suggestions for Authors

Review for the manuscript “Long non-coding RNA signatures in the ileum and colon of  Crohn´s disease patients and effect of anti-TNF-α treatment on their modulation”.

Overall comments: This study performed transcriptomic profiling of lncRNA by RNA sequencing to characterize lncRNA signatures in the inflamed and non-inflamed ileum and colon from CD patients. Moreover, the authors investigated the ex vivo effect of anti-TNF-α on intestinal mucosa. The quality of this study is very good.

ABSTRACT

            This section is adequate.

KEYWORDS

            I suggest including the definition of lncRNA among the keywords.

INTRODUCTION

            This section is adequate. However, I miss the inclusion of newer references. I suggest including more references published in 2022 and 2023. The authors can find these references in different databases such as PUBMED and EMBASE.

METHODS

            In lines 88-92, we can read “…all of them had macroscopically and histologically normal mucosa. This study was approved by the local ethics committee and written informed consent was obtained from all the participants prior to sample collection. Main demographic characteristics of CD patients (aCD and qCD) and HC are shown in Table 1.” Please include the name of the Ethics Committee and the date of the project's approval.

RESULTS

            The quality of Figure 1 should be improved. The same is valid for Figures 2-4, mainly for Figures 3 and 4.

DISCUSSION

            This section is adequately performed; however, At its beginning, we can find “The present study aims to characterize lncRNA signatures in the ileum and colon of CD patients to elucidate the molecular mechanisms involved in the pathogenesis of CD as a starting point for future therapeutic strategy development. Moreover, we have analyzed the potential role of anti-TNF-α agents in the modulation of lncRNA expression on intestinal mucosa using an ex vivo approach.” I do not see reasons to repeat the study's aims in this section. I suggest removing the contents found in lines 230-234.

            Please use italics for “in vivo”, “in vitro” and "ex vivo”.

REFERENCES

            As pointed out, I suggest including newer references in the Introduction section.

Comments on the Quality of English Language

Minor revisions are required.

Author Response

Response to Reviewer 3 Comments

This study performed transcriptomic profiling of lncRNA by RNA sequencing to characterize lncRNA signatures in the inflamed and non-inflamed ileum and colon from CD patients. Moreover, the authors investigated the ex vivo effect of anti-TNF-α on intestinal mucosa. The quality of this study is very good.

Point 1.    I suggest including the definition of lncRNA among the keywords

Response 1. We have considered your comments and we have included the definition of lncRNA among the keywords.

Point 2.  This section is adequate. However, I miss the inclusion of newer references. I suggest including more references published in 2022 and 2023. The authors can find these references in different databases such as PUBMED and EMBASE

Response 2. Agree. We have included more references published in 2022 and 2023.

Point 3.  In lines 88-92, we can read “…all of them had macroscopically and histologically normal mucosa. This study was approved by the local ethics committee and written informed consent was obtained from all the participants prior to sample collection. Main demographic characteristics of CD patients (aCD and qCD) and HC are shown in Table 1.” Please include the name of the Ethics Committee and the date of the project's approval.

Response 3. We have considered your comments and we have included the name of the Ethics Committee and date of approval.

Point 4.  The quality of Figure 1 should be improved. The same is valid for Figures 2-4, mainly for Figures 3 and 4.

Response 4. Agree. We have improved the quality of the figures so that they can be better visualized.

Point 5. This section is adequately performed; however, At its beginning, we can find “The present study aims to characterize lncRNA signatures in the ileum and colon of CD patients to elucidate the molecular mechanisms involved in the pathogenesis of CD as a starting point for future therapeutic strategy development. Moreover, we have analyzed the potential role of anti-TNF-α agents in the modulation of lncRNA expression on intestinal mucosa using an ex vivo approach.” I do not see reasons to repeat the study's aims in this section. I suggest removing the contents found in lines 230-234.

Please use italics for “in vivo”, “in vitro” and "ex vivo”.

Response 5. We have considered your comments and we have removed the first paragraph (lines 230-234) in the new version. In addition, we have used italics for in vitro and in vivo.

Point 6. As pointed out, I suggest including newer references in the Introduction section

Response 6. Agree. We have included newer references published in 2022 and 2023.

Reviewer 4 Report

Comments and Suggestions for Authors

This is a very interesting and original study that open new routes for better selection of CD patients for treatment with biologic therapies

Their  results provide an in-depth analysis of the lncRNA landscape in the ileum and colon of patients with CD (active and quiescent) and HC.

In addition, the level of expression of some transcripts was associated with the presence of inflammation in CD.

Given the lack of knowledge on the role of lncRNAs in the pathogenesis of CD, their location-specific expression profile might lay a foundation for future research to investigate potential therapeutic targets in ileal and colonic CD.

Author Response

Response: We thank the reviewer for his comments.